# Block Copolymers of Poly(N-Vinyl Pyrrolidone) and Poly(Vinyl Esters) Bearing n-alkyl Side Groups via Reversible Addition-Fragmentation Chain-Transfer Polymerization: Synthesis, Characterization, and Thermal Properties

**DOI:** 10.3390/polym16172447

**Published:** 2024-08-29

**Authors:** Nikoletta Roka, Theodosia-Panagiota Papazoglou, Marinos Pitsikalis

**Affiliations:** Industrial Chemistry Laboratory, Department of Chemistry, National and Kapodistrian University of Athens, Panepistimiopolis Zografou, 15771 Athens, Greece; nikolettaroka@yahoo.gr (N.R.); pnpapazoglou@gmail.com (T.-P.P.)

**Keywords:** poly(N-vinyl pyrrolidone) (NVP), poly(vinyl esters) (VEs), block copolymers, RAFT polymerization, thermal analysis, thermal stability

## Abstract

Amphiphilic block copolymers of N-vinyl pyrrolidone (NVP) and various vinyl esters (VEs), PNVP-*b*-PVEs, namely vinyl butyrate (VBu), vinyl decanoate (VDc), and vinyl stearate (VSt), were synthesized through RAFT polymerization techniques. The sequential addition of the monomers methodology was employed starting from the polymerization of NVP followed by the polymerization of the Ves’ monomer. The polymerization of NVP was conducted at 60 °C in benzene solution using AIBN as the initiator and O-ethyl S-(phthalimidylmethyl) xanthate as the CTA. The resulting PNVP macro-CTA was further applied for the polymerization of the vinyl ester in dioxane solution at 80 °C using, again, AIBN as the initiator. The block copolymers were characterized through size-exclusion chromatography (SEC) and NMR spectroscopy. The thermal behavior of the copolymers was studied by Differential Scanning Calorimetry (DSC), whereas their thermal stability via Thermogravimetric Analysis (TGA) and Differential Thermogravimetry (DTG).

## 1. Introduction

Poly(vinyl esters) (PVEs) consist of a valuable class of polymeric materials [1], mainly for two reasons. The first one has to do with the numerous applications of these materials in the industrial sector, including their employment as elastomers, plastics, fibers, coatings, paints, additives, adhesives, textiles, cosmetics, etc. [2,3,4,5,6,7]. This broad range of applications is associated with the correlation between the structure and the properties of the PVEs. Depending on the nature of the side ester group (aliphatic with various carbon atoms, aromatic, alicyclic, olefinic, etc.), the PVEs can be either hydrophilic or hydrophobic, amorphous or semi-crystalline, liquid-like, waxy, or solid, etc. [8,9,10,11,12,13,14,15,16,17,18]. The second reason has to do with the fact that PVEs can be precursors for the production of poly(vinyl alcohol) (PVA), which is a benchmark water-soluble polymer demonstrating non-toxicity, non-carcinogenic properties, and possibilities for bio-conjugation for therapeutical applications [19,20,21]. Therefore, it can be applied as a valuable biomaterial in pharmacological applications [22,23,24]. The transformation of PVEs into PVA is conducted through hydrolysis of the ester group. The nature of the side group affects the kinetics of hydrolysis, ranging from a few minutes to several hours. Under these conditions, intermediate degrees of hydrolysis can be reached, leading to products with complex behavior in aqueous solutions [25].

The most well-known member of the PVE polymer family is poly(vinyl acetate) (PVAc). It is the most widely used precursor for the synthesis of PVA, and numerous studies have been devoted to the synthesis of this material and its copolymers with other VEs or other monomers [26,27]. Much less work has been devoted to the synthesis and the study of the solution and solid-state properties of other members of the PVE family of polymers [28,29,30,31,32].

The polymerization of VEs is feasible only through radical polymerization. Conventional radical polymerization, although straightforward in its applications, suffers various disadvantages (lack of control over the molecular weights, high polydispersity, Ð, values, inability to produce pure end-functionalized polymers, very limited possibilities to synthesize complex macromolecular architectures, etc.) [33]. In particular, for VAc, extended studies have been performed to reveal the presence of chain-transfer reactions to monomers and polymers, leading to the formation of branched structures and the appearance of regio-irregularities with increased head-to-head linkages [34,35,36,37]. For this reason, the polymerization of VEs was not a hot topic in polymer chemistry for many years. However, developments in controlled radical polymerization methodologies opened new horizons for the controlled polymerization of VEs and offered the possibility to manipulate macromolecular engineering towards the synthesis of tailor-made polymers. Among these approaches, nitroxide-mediated, atom-transfer-radical, iodine-transfer, organostibine-mediated, vanadium-mediated, and cobalt-mediated polymerization techniques have been employed [38,39,40,41,42,43,44,45]. Although relative success has been reported in several cases, these methods present several drawbacks, rendering their universal application problematic. By far the most successful polymerization technique for VEs is the Reversible Addition−Fragmentation Chain-Transfer (RAFT) methodology, as it allows for the synthesis of controlled structures in terms of both the molecular characteristics and the copolymer topology or macromolecular architecture in combination with other polymerization techniques and a variety of monomers [46,47].

In this work, the synthesis of amphiphilic block copolymers of N-vinyl pyrrolidone (NVP) and various vinyl esters, PNVP-*b*-PVEs, including vinyl butyrate (VBu), vinyl decanoate (VDc), and vinyl stearate (VSt), is described through RAFT approaches. These materials are expected to provide very interesting properties. On the one hand, PNVP is an amorphous polymer with a high Tg value (equal to 187 °C, depending on the humidity) that is soluble in both aqueous solutions and in organic solvents without showing lower critical solution temperature (LCST) [48,49,50,51,52,53]. On the other hand, PVBu and PVDc are also amorphous polymers with low Tg values (−5 °C for PVBu and −45 °C for PVDc), whereas PVSt is a semicrystalline polymer with Tm = 52–57 °C [54,55,56]. In the literature, several efforts have been presented for the synthesis of statistical and block copolymers of different VEs, including VAc in all cases and other monomers from the same family, such as vinyl pivalate, vinyl benzoate, vinyl octanoate, etc. In the past, the corresponding block copolymers of NVP with VAc have been frequently reported in the literature. However, very limited work has been reported that involves the synthesis of block copolymers of NVP with other VEs.

All monomers used in this study, NVP and VEs, are considered to belong to the category of less-activated monomers in RAFT polymerization. The monomers are classified into two families in this technique of polymerization. They are characterized either as more-activated or less-activated monomers. The key parameter differentiating these two groups of monomers is their ability to stabilize radicals [47,57]. In the present work, the difference in reactivity between NVP and VEs is considered to be low, thus making it easier to combine these monomers into a single polymerization system and thus providing better control over the copolymerization procedure. However, difficulties exist in the specific system of monomers. VAc is considered to be the least active monomer among the common LAMS in RAFT polymerization because it provides highly reactive and thus unstable propagating radicals. Therefore, it is not very easy to control the polymerization reaction. The situation becomes even more problematic upon increasing the size of the n-alkyl side group of the VEs, leading to extended retardation of the polymerization reaction and the possibility to have pronounced termination reactions, rendering control of the macromolecular architecture even more difficult [58,59,60].

## 2. Materials and Methods

### 2.1. Materials

N-Vinyl pyrrolidone (≥97% FLUKA, Csomád, Hungary) containing sodium hydroxide as the inhibitor was dried overnight over calcium hydride and distilled prior to use. The vinyl esters (TCI Chemicals, Chennai, India, VBu > 98%, VDc > 99%, and VSt > 95%) stabilized with monomethylether hydroquinone were also dried over calcium hydride overnight and then distilled under vacuum prior to polymerization. VBu has a bp of 115–117 °C, whereas VDc has a bp of 119–120 °C. On the other hand, VSt is a solid monomer, and it was purified after dissolution in THF and being passed through an inhibitor-remover column. Azobisisobutyronitrile AIBN (98% ACROS, Geel, Belgium) was purified through recrystallization twice from methanol and then dried under vacuum. The Chain-Transfer Agent, O-ethyl S-(phthalimidylmethyl) xanthate, was synthesized according to the literature protocols [61,62] by employing O-Ethyl xanthic acid potassium salt and N-(bromomethyl)phthalimide. Dioxane and benzene were also purified over CaH_2_ overnight and distilled just prior to use.

All other reagents and solvents were of commercial grade and used as received.

### 2.2. Synthesis of PNVP-b-PVE Block Copolymers via RAFT Polymerization

The synthesis of the PNVP-*b*-PVE block copolymers was accomplished through the sequential addition of monomers, starting from the polymerization of NVP. O–ethyl S–(phthalimidymethyl) xanthate was employed as the CTA and AIBN as the initiator. The polymerization of NVP was conducted in glass reactors using high-vacuum techniques [63,64] at 60 °C in benzene solutions for 12 h.

A typical polymerization procedure for NVP with final M_n_ = 8.9 × 10^3^ (Table 1, samples PNVP-*b*-PVBu #3 and #4) was accomplished using a molar ratio of [NVP]_0_/[CTA]_0_/[AIBN]_0_ = 100/1/0.2, and it is described as follows: 5 g of NVP was polymerized in the presence of 0.1284 g of CTA and 0.0148 g of AIBN in 5 mL of benzene. The polymerization solution was subjected to three freeze–thaw pump cycles in order to eliminate the oxygen from the polymerization apparatus. The reactor was flame-sealed and placed in a preheated oil bath at 60 °C for 12 h.

The reaction was terminated by removing the reaction flask from the oil bath and through immediate cooling of the polymerization mixture under the flow of cold water. The apparatus was then opened, thus exposing the mixture to air. The polymer was then precipitated in an excess of diethyl ether. This procedure was repeated at least three times in order to ensure the removal of any unreacted monomer residues. The polymers were finally dried overnight in a vacuum oven at 50 °C to remove any residual solvent. The conversions for all homopolymers were near quantitative.

The PNVP homopolymers served as the macro-CTAs to promote the polymerization of the VEs in a subsequent step. The block copolymerization reactions were performed in dioxane solutions at 80 °C for 96 h in glass reactors under high-vacuum conditions, as previously reported for the synthesis of the PNVP macro-CTAs. The quantities of the VE monomers, the AIBN radical initiator, and the dioxane solvent employed for the synthesis of the block copolymers are reported in Table 1. The polymerization mixture was subjected to three freeze–thaw pump cycles in order to eliminate the oxygen from the polymerization apparatus. The polymerization was terminated by removing the reactor from the oil bath and cooling the mixture under a flow of cold water. The reactor was then opened, and the copolymerization mixture was exposed to air. The copolymers with the PVBu blocks were precipitated in an excess of hexanes, whereas the copolymers with PVDc and PVSt blocks were precipitated in methanol. The crude product was dissolved in THF and reprecipitated in the appropriate non-solvent (hexanes or methanol). This procedure was repeated three times in order to ensure the removal of any unreacted monomer residues. Finally, the copolymers were dried overnight in a vacuum oven at 50 °C to remove any residual solvent. No further effort was needed to purify the samples.

### 2.3. Characterization Techniques

The molecular weight (M_w_) as well as the molecular weight distribution, Ð = M_w_/M_n_, were determined through size-exclusion chromatography, SEC, by employing a modular instrument consisting of a Waters model 510 pump, a U6K sample injector, a 401 differential refractometer, and a set of 5 μ Styragel columns with a continuous porosity range from 500 to 10^6^ Å. The carrier solvent was CHCl_3_, and the flow rate 1 mL/min. The system was calibrated using nine polystyrene standards with molecular weights in the range of 970–600,000.

The composition of the copolymers was determined from their ^1^H NMR spectra, which were recorded in chloroform-d at 30 °C with a 400 MHz Bruker Avance Neo spectrometer (Billerica, MA, USA).

The T_g_ values of the copolymers were determined using a 2910 Modulated DSC Model from TA Instruments. The samples were heated under a nitrogen atmosphere at a rate of 10 °C/min from −50 °C to 220 °C. The second heating results were obtained in all cases.

The thermal stability of the copolymers was studied through Thermogravimetric Analysis (TGA) by employing a Q50 TGA model from TA Instruments. The samples were placed in a platinum pan and heated from ambient temperatures to 600 °C in a 60 mL/min flow of nitrogen at heating rates of 10 °C/min. 

## 3. Results and Discussion

### 3.1. Synthesis of the Block Copolymers of NVP and VEs

The synthesis of the desired block copolymers was accomplished via RAFT polymerization and the sequential addition of monomers. Because both types of monomers have low reactivities, the same CTA can be employed for the synthesis of well-defined block copolymers [65,66,67,68]. Several families of CTAs have been applied for the polymerization of LAMs in RAFT polymerization, including xanthates and dithiocarbamates. In the present study, O-ethyl S-(phthalimidylmethyl) xanthate was used as the CTA, as it is well-known that it promotes very good control over the RAFT polymerization of LAMs. In particular, it has been efficiently employed for the homopolymerization of both NVP and VEs in the past. Specifically, VAc was polymerized with this CTA, leading to controlled molecular weights covering a range of values as low as 3000 up to 121,000 with relatively low polydispersities, Ð, in the range of 1.2 to 1.4. An induction period of about 1 h was revealed in kinetic experiments upon performing the polymerization reaction at 60 °C [69,70,71].

The procedure for the synthesis of the PNVP-*b*-PVE block copolymers is depicted in Figure 1.

NVP was polymerized first, providing well-defined macromolecular CTAs (macro-CTAs) with controlled molecular characteristics, i.e., relatively narrow molecular weight distribution and molecular weight, which was close to the stoichiometric values. In order to better control the dispersity values and to achieve quantitative incorporation of the CTAs’ moieties as the polymers’ end-groups, the polymerization was not allowed to proceed to very high yields. In most cases, the reaction was terminated by cooling the reaction flask when the conversion was less than 80% in all cases. At higher yields, the solution becomes very viscous, and the system becomes more or less heterogeneous, leading to increased dispersity values. In addition, upon increasing the polymerization yield, the possibility of losing control over the end-groups is increased. The PNVP macro-CTAs were precipitated, re-dissolved, and re-precipitated to remove unreacted monomer. The samples were finally dried in a vacuum oven. 

The purified and dried macro-CTAs were further employed for the polymerization of various VEs, namely VBu, VDc, and VSt. In order to minimize the viscosity problems, the termination, and other side reactions involved in RAFT radical polymerization, thus achieving the best degree of control during the copolymerization reaction, the conversion of the polymerization of VEs was not allowed to reach high values, i.e., less than 50% in most cases. The efficiency of this approach was monitored through SEC and ^1^H NMR measurements. Characteristic SEC traces from the synthesis of the block copolymers are provided in Figure 1, whereas a typical ^1^H NMR spectrum is shown in Figure 2. More data are available in the Appendix A. Table 2 incorporates the molecular characteristics of all of the synthesized block copolymers.

The SEC traces indicate for all block copolymers that the polymerization of the second monomer was efficiently promoted by the macro-CTA. Single peaks with only minor tailing effects in a few cases were obtained, showing that it is impossible to completely avoid termination phenomena, something that is already established in RAFT polymerization [46]. Consequently, no effort was made to further purify the block copolymers. The reverse mode of monomer addition was not tested, as it is known that the chain extension employing PVEs’ macro-RAFT agents presents difficulties due to the instability of the PVEs’ macroradicals. The safer way to minimize chemical heterogeneity in block copolymer synthesis and achieve lower polydispersity values involves the synthesis of the PNVP macro-RAFT agent initially and then the polymerization of the less-reactive VEs.

The efficiency of the synthetic procedure was further manifested by the ^1^H NMR analysis. The characteristic signals of both monomer units are obvious in the spectra of the copolymers. The signal at 3.2 ppm, which is attributed to the methylene protons of the pyrrolidone ring that are adjacent to the nitrogen atom, was employed for the calculation of the composition of the PNVP block. On the other hand, for the PVEs’ blocks, the signals at 0.9 ppm, attributed to the methyl end-groups of the side groups of the PVEs, were used for the calculation of the content in PVEs. The results, given in Table 2, reveal that it is possible to manipulate the composition of the block copolymers from the synthetic route.

### 3.2. Thermal Properties

#### 3.2.1. DSC Analysis

The thermal properties of the block copolymers were studied through DSC measurements. The results for the PNVP-*b*-PVBu copolymers are given in Table 3, whereas the DSC thermograms are provided in Figure 3.

Both components of the blocks are amorphous. Their Tg values are located at 187.1 °C [52,53] and −5 °C [72] for the PNVP and PVBu homopolymers, respectively. In most cases, three different Tg values are obvious from the thermograms. The high Tg value is relatively close to the Tg value of the PNVP homopolymer, although it is considerably lower. The low Tg value is close to that of the PVBu homopolymers. It has been reported that the Tg value for PVBu is −5 °C, which is slightly higher than the one reported in this work for the corresponding homopolymer. This small difference is attributed to the lower-molecular-weight sample employed in the present study. Finally, the third Tg value is located at intermediate temperatures between those of the two homopolymers. These results clearly indicate that the block copolymers are microphase-separated. However, due to the relatively low molecular weight of the copolymers, the samples are not located at the strong segregation limit [73]. The third intermediate transition confirms a partial mixing of the two phases. This mixing of the two phases is so substantial that it acts as a third phase in the system, leading to the appearance of a third Tg from the created mesophase. Only the sample PNVP-*b*-PVBu #2 shows the absence of a pure PVBu phase, as this sample has the lower content in PVBu (16% mol according to NMR analysis). The results confirm that there is an almost pure PNVP microphase (indicated by the highest Tg value, which is relatively close to the corresponding homopolymer) accompanied by a second microphase from the mixing of the PVBu and PNVP components. Sample #1 with the lowest PNVP content (22% mol in PNVP according to NMR analysis) has the lowest Tg value for the PNVP microphase, as this phase is “contaminated” by the highest amount of PVBu. Similar microphase separation data obtained through DSC measurements have appeared in the literature for block copolymers of different PVEs [54,55].

The DSC data for the PNVP-*b*-PVDc block copolymers are illustrated in Table 4, whereas characteristic thermograms are provided in Figure 4.

It is obvious that the Tg value of PVDc [54] is much lower than that of PVBu, as the Tg value drastically decreases as the side chain of the main polymer increases [74,75,76,77]. This result is reasonable, and it is attributed to an internal plasticization effect, which is well-documented in polymer chains carrying aliphatic side chains. The thermal behavior is similar to that observed in the case of the copolymers PNVP-*b*-PVBu. For all of the samples, three transitions are observed, indicating the existence of microphase separation in the copolymers. However, due to the rather low molecular weights of the copolymers, the system is at the low segregation limit, with partial mixing of the phases. The difference between the copolymers with PVDc bocks compared to those with the PVBu blocks is that even with very low PVDc content (as low as 7% mol in sample PNVP-*b*-PVDc #4), the low Tg phase of the PVDc component is clearly observed (even with a much higher Tg value due to contamination with PNVP chains). This is direct evidence that the PNVP-*b*-PVDc copolymers have a greater tendency towards microphase separation than the PNVP-*b*-PVBu copolymers, or, in other words, that the Flory–Huggins χ parameter is much higher between the PNVP and PVDc chains than the PNVP and PVBu chains. Consequently, the increase in the size of the side-chain in PVEs facilitates the microphase-separation procedure.

The thermal transitions of the PNVP-*b*-PVSt block copolymers are summarized in Table 5, whereas the DSC traces are shown in Figure 5.

In this case, the situation is more complex, as the homopolymer PVSt is a semicrystalline polymer showing side-chain crystallization due to the extended side-chains of 18 carbon atoms [55,72]. In similar families of polymers bearing long linear alkyl side groups, such as methacrylates, acrylates, and α-olefins, it has been shown that when the number of carbon atoms of the linear side group is higher than 12, side-chain crystallization occurs [74,75,76,77]. The experimental data, shown in Figure 5, confirm the presence of the melting endotherm during heating for all samples. The Tm value for the PVSt homopolymer has been reported in the range of 52 up to 57 °C [14]. The lower Tm value in this study has to do with the relatively low molecular weight of the homopolymer used in this work. However, the enthalpy of melting agrees very well with the data from the literature and is equal to 87.9 J/g. Compared to the melting point, the Tm of the PVSt homopolymer, the Tm values of the copolymers are located at lower temperatures due to the fact that the crystallization procedure is restricted by the presence of the amorphous block of the PNVP chains. Similar results have been reported in numerous other cases of block copolymers bearing an amorphous and a semicrystalline block. It is characteristic that the lower Tm value was obtained in the sample PNVP-*b*-PVSt #1, which has the lower molecular weight of the PVSt block.

The case of diblock copolymers consisting of a crystallizable block and another amorphous block has been previously examined theoretically by Flory [78]. A lattice model was applied to analyze the thermodynamics of crystallization of homopolymers and copolymers. In the case of the copolymers, a considerable decrease in the melting point of the crystallizable block compared to the pure homopolymer was predicted. Both the length of the crystallizable block and the thermodynamic interactions between the two blocks play a crucial role in affecting the melting-point depression.

The Tg of the PVSt, which is expected to be very low (as low as −100 °C), was not observed experimentally, as it is a weak transition due to the crystallinity of the polymer. From the results shown in Table 6, it is obvious that the samples are microphase-separated due to the observation of both the PVSt melting points and the PNVP Tg values, which are very close to the expected ones considering the molecular weight dependence of the Tg of PNVP. This result is further supported by the data obtained by Mayes, Russell, et al. [79] concerning the phase behavior of a similar system of block copolymers consisting of polystyrene and poly(n-alkyl methacrylates) with up to 12 carbon atoms. It was found that the interaction parameter, χ, increases by increasing the length of the alkyl side group of the polymethacrylate block.

The degrees of crystallinity were calculated by measuring the normalized enthalpy of melting of the PVSt blocks (ΔH, j/PVStg) in the copolymers compared to the same value for the 100% crystalline homopolymer (ΔHo = 220 j/PVStg) [55], as follows:Xc = ΔH (j/PVStg)/ΔHo (j/PVStg)

The results, shown in Table 5, reveal that the copolymers have relatively low degrees of crystallization (less than 20% in all cases), showing that the presence of the PNVP blocks restricts the organization of the crystalline phases and that the more significant part of the PVSt block remains amorphous.

Taking these results into account, it is reasonable to expect that the amorphous phase of the PVSt will be responsible for the appearance of a second Tg value, except for that attributed to the amorphous PNVP phases. This hypothesis was proven true by the data presented in Table 5 and the thermograms given in Figure 5. The relative values of the two transitions confirm that the two amorphous phases are partially mixed due to the low molecular weights of the constituent blocks. Finally, in the case of the PNVP-b-PVSt block copolymers, a crystalline phase of the PVSt block and two mixed phases of the amorphous components are present.

#### 3.2.2. TGA Analysis

The thermal stability of the PNVP-*b*-PVE block copolymers was studied through TGA and DTG analysis. The results for the PNVP-*b*-PVBu copolymers are given in Table 6 and the TGA plots in Figure 6. The corresponding DTG plots are given in the Appendix A.

The PNVP homopolymer is a relatively thermally stable polymer showing a single decomposition peak based on the DTG analysis [80]. The maximum temperature of the thermal decomposition peak is located at 437 °C. These results indicate a rather simple decomposition mechanism. On the other hand, the PVBu homopolymer reveals a two-step thermal decomposition mechanism with two maxima located at 319 and 416 °C, respectively. PVBu is a much less thermally stable polymer compared to PNVP, as the main decomposition procedure is the one that takes place at the lower decomposition temperature [81,82].

All of the PNVP-b-PVBu block copolymers revealed two steps of thermal decomposition, indicating similar complex decomposition mechanisms, as in the case of the PVBu homopolymer. The first thermal decomposition peak was traced at temperatures close to the low-temperature decomposition peak of the PVBu homopolymer, and it is obviously correlated to the decomposition of the PVBu block. The second thermal decomposition is located at higher temperatures, and it is associated with the decomposition of the PNVP block. This conclusion was further confirmed by comparing the mass loss with the composition of the block copolymers. The first decomposition peak of sample #1, which has the highest PBVu content, is the most pronounced among the other samples. On the contrary, in the case of sample #2, which has the highest PNVP content, the high-temperature decomposition peak is very pronounced. The other two samples with similar compositions also have a similar decomposition pattern.

The TGA results for the PNVP-b-PVDc copolymers are given in Table 7, whereas the TGA plots are in Figure 7. The corresponding DTG plots are given in the Appendix A.

The thermal decomposition of PVDc is similar to that of PVBu. It takes place at two distinct steps with maxima at 323 °C and 418 °C, respectively. As in the case of PVBu, the first decomposition step is the most important, corresponding to more than 80% of mass loss of the initial sample. The increase in the size of the n-alkyl side group does not appreciably increase the thermal stability of the polymer chain. Two thermal degradation steps were also observed for the PNVP-*b*-PVDc blocks. The low-temperature peak corresponds to the decomposition of the PVDc component, whereas the high-temperature step corresponds to the PNVP component. As in the case of the PNVP-*b*-PVBu copolymers, the composition of the PNVP-*b*-PVDc is reflected in the relative size of the two decomposition peaks in the DTG analysis.

The TGA results for the PNVP-*b*-PVSt copolymers are given in Table 8, whereas the TGA plots are in Figure 8. The corresponding DTG plots are given in the Appendix A. The thermal decomposition of PVSt is substantially different compared to that reported for PVBu and PVDc. The main decomposition event is observed at 320 °C, as in the other polymers. However, a considerably important degradation step, corresponding to about 40% of mass loss, is obvious at much lower temperatures, around 195 °C. In addition, a shoulder at temperatures higher than 400 °C is visible, indicating a complex mechanism of thermal degradation.

Taking into account that the degradation step at 320 °C is more or less common to all PVEs, this should be attributed to the decomposition of the main poly(vinyl ester) backbone, whereas the lower-temperature peak can be attributed to the thermal decomposition of the side n-alkyl group of PVSt.

All of the PNVP-b-PVSt copolymers showed three-step degradation patterns. The high-temperature step is attributed to the thermally more stable PNVP block, whereas the other peaks from the DTG analysis correspond to the less thermally stable PVSt block. As in the previous cases, the composition of the copolymers is reflected in the relative size of the various peaks.

The thermal decomposition of several members of the PVE family has been thoroughly studied in the literature [81,82]. An autocatalytic degradation mechanism takes place for the first members of the PVEs, especially PVAc. The higher members of this family of polymers degrade through first-order kinetics. A combination of autocatalytic and first-order kinetics approaches seems to exist for the intermediate members of PVEs; therefore, both mechanisms have been examined in these cases. A two-step decomposition process was revealed for most PVEs. The first one involves the removal of the ester group as an acid and then the decomposition of the remaining chain. These conclusions were more or less verified in the present study.

## 4. Conclusions

Well-defined block copolymers of N-vinyl pyrrolidone (NVP) and various vinyl esters (VEs) with aliphatic side groups, namely, vinyl butyrate (VBu), vinyl decanoate (VDc), and vinyl stearate (VSt), were synthesized through RAFT polymerization techniques. The sequential addition of monomers was employed, starting from the polymerization of NVP in benzene solutions at 60 °C using AIBN as the radical initiator and O-ethyl S-(phthalimidylmethyl) xanthate as the CTA. The polymerization of the VEs was conducted in dioxane solutions at 80 °C by using the PNVP block as the macro-CTA and AIBN as the initiator. Block copolymers of various compositions were obtained. Size-exclusion chromatography (SEC) analysis revealed that products of relatively low polydispersity were obtained and that there was no reason for purification of the block copolymers. Differential scanning calorimetry (DSC) analysis showed that the PNVP-b-PVBu and PNVP-b-PVDc block copolymers were microphase-separated. However, due to the rather low molecular weight of the samples, partial mixing of the two phases was observed. On the other hand, the PNVP-b-PVSt block copolymers were semicrystalline. The samples were again microphase-separated, as both the crystalline phase of the PVSt blocks and the amorphous phase of the PNVP blocks were traced in the thermograms. A third amorphous phase, due to the partial mixing of the two constituent blocks, was also obtained. The thermal stability of the copolymers was studied via Thermogravimetric Analysis (TGA) and Differential Thermogravimetry (DTG). PNVP was a considerably more thermally stable polymer than the PVEs. A rather simple mechanism of thermal decomposition was applied for PNVP, whereas at least two steps of thermal degradation were traced for the various PVE homopolymers. The block copolymers combined the thermal decomposition behaviors of their constituent components, thus showing a complex pattern of thermal degradation.

## Data Availability

The data are available upon request.

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
