# Peer review of "Block Copolymers of Poly(N-Vinyl Pyrrolidone) and Poly(Vinyl Esters) Bearing n-alkyl Side Groups via Reversible Addition-Fragmentation Chain-Transfer Polymerization: Synthesis, Characterization, and Thermal Properties"

_polymers, 2024, doi:10.3390/polym16172447_

Round 1
Reviewer 1 Report
Comments and Suggestions for Authors
Pitsikalis and coworkers reported the synthesis of Poly(N-vinyl Pyrrolidone)-block-Poly(Vinyl Esters) with varying alkyl side lengths. The authors tried to provide details on the synthesis and thermal characteristics of the block copolymers by DSC, TGA, and DTG, but I think the manuscript can be much improved for the publications.
1. Mentioning specific conditions (e.g. dioxane 80oC) in the abstract is untypical and unnecessary. This can be misleading considering that the first block PNVP is synthesized in benzene at 60oC. The polymerization condition of dioxane 80oC cannot be generalized for the whole synthetic process.
2. My major comment on the manuscript is to consider making supporting information. The manuscript at current state simply lists too many figures and tables (~14 figures and 8 tables). This is redundant, makes the manuscript seem unorganized and hard to read. Please highlight the major findings and important data in the main text, and move everything else to supporting information so that the manuscript can be focused and look professional. Maybe experimental section can be also moved to the SI if the editorial allows?
3. In the introduction section, 3rd paragraph and last paragraph are repetitive. I think 3rd paragraph is unnecessary and can be combined with the last paragraph to better address and summarize this work.
4. I see unnecessary abbreviations: less activated monomers (LAMs), and more activated monomers (MAMs). They only appear a few times in the introduction section and aren’t worth an abbreviation.
5. Inconsistent capitalization:
Abstract: SEC, DSC, TGA, DTG
Page 2 line 12 : Polymer Chemistry / Line 20: RAFT
For example, DSC, TGA, etc., these are typical abreviations/capitalizations and are acceptable. However, I think capitalization in the full name seems weird.
Also, Polymer Chemistry is an area of research. If authors tried to emphasize this field, may be acceptable. However, this is not in line with other abbreviations mentioned above.
6. Based on the description in the experimental section, it doesn’t seem that the authors are doing block copolymerization by sequential addition of monomers. This can be very misleading. Readers including myself will think that the authors do block copolymerization in one-pot by sequential addition of monomers in the same batch, which is not the case in this work.
7. In experimental 2.2 section, please specify the ‘appropriate non-solvent’ used for the reprecipitations.
8. Peak signals for the Tgs in DSC curves are very weak in Fig 7 and 8. I’m worried if the peak identification was good enough. Also, the authors simply state that 3 Tgs observed in the block copolymers are from each block and the microphase-separated structure. Authors need to provide supporting details for the microphase separation – or may be at lease an appropriate references.
Comments on the Quality of English Language
-
Author Response
We are thankful to the Reviewer for the fruitful comments and suggestions. Everything was taken into account and the manuscript was revised appropriately. The changes in the text are given by red color. Our point-by-point answers to the Reviewer are given below:
Pitsikalis and coworkers reported the synthesis of Poly(N-vinyl Pyrrolidone)-block-Poly(Vinyl Esters) with varying alkyl side lengths. The authors tried to provide details on the synthesis and thermal characteristics of the block copolymers by DSC, TGA, and DTG, but I think the manuscript can be much improved for the publications.
- Mentioning specific conditions (e.g. dioxane 80oC) in the abstract is untypical and unnecessary. This can be misleading considering that the first block PNVP is synthesized in benzene at 60o The polymerization condition of dioxane 80oC cannot be generalized for the whole synthetic process.
We agree with the Reviewer and therefore the text was revised accordingly.
- My major comment on the manuscript is to consider making supporting information. The manuscript at current state simply lists too many figures and tables (~14 figures and 8 tables). This is redundant, makes the manuscript seem unorganized and hard to read. Please highlight the major findings and important data in the main text, and move everything else to supporting information so that the manuscript can be focused and look professional. Maybe experimental section can be also moved to the SI if the editorial allows?
We rearranged the manuscript as the Reviewer suggested. Several figures were removed to the Supporting Information Section, SIS. However, we believe that the experimental section cannot be moved to the SIS, since important information is included therein and in addition it is not very expanded describing very briefly the experimental procedures.
- In the introduction section, 3rdparagraph and last paragraph are repetitive. I think 3rd paragraph is unnecessary and can be combined with the last paragraph to better address and summarize this work.
The 3rd paragraph cannot be completely eliminated, since it briefly describes how we moved from the conventional radical polymerization of VEs to RAFT polymerization. However, this paragraph was shortened. The last paragraph of the introduction section was also shortened and partially mixed with the 4th paragraph.
- I see unnecessary abbreviations: less activated monomers (LAMs), and more activated monomers (MAMs). They only appear a few times in the introduction section and aren’t worth an abbreviation.
The abbreviations LAMs and MAMs are very common in RAFT polymerization and that is why they were referred to the original text. However, we understand the concern of the Reviewer and we deleted these abbreviations.
- Inconsistent capitalization:
Abstract: SEC, DSC, TGA, DTG
Page 2 line 12: Polymer Chemistry / Line 20: RAFT
For example, DSC, TGA, etc., these are typical abbreviations/capitalizations and are acceptable. However, I think capitalization in the full name seems weird.
Also, Polymer Chemistry is an area of research. If authors tried to emphasize this field, may be acceptable. However, this is not in line with other abbreviations mentioned above.
The typical and well-known abbreviations SEC, DSC, TGA, DTG and RAFT were left as mentioned in the text. However, capitalization in other instances were avoided, as suggested.
- Based on the description in the experimental section, it doesn’t seem that the authors are doing block copolymerization by sequential addition of monomers. This can be very misleading. Readers including myself will think that the authors do block copolymerization in one-pot by sequential addition of monomers in the same batch, which is not the case in this work.
It is clearly reported in the text that we do not have a one-pot sequential addition of monomers. It is obvious that the polymerization of NVP is not allowed to be quantitative to ensure the presence of the desired end-groups that will make the resulting PNVP block efficient macro-CTA for the polymerization of the VEs. We had to remove the excess of unreacted monomer after the formation of the first block. The term “sequential addition of monomers” is employed to point out that the polymerization of the second monomer takes place without the chemical transformation of the first block. The originally prepared PNVP chains serve directly as the macro-CTA for the efficient RAFT polymerization of the VEs.
- In experimental 2.2 section, please specify the ‘appropriate non-solvent’ used for the reprecipitations.
The copolymers with the PVBu blocks were precipitated in excess of hexanes, whereas the copolymers with PVDc and PVSt blocks in methanol. The crude products were redissolved in THF and reprecipitated in the appropriate non-solvent, which is either hexanes (for the blocks with PVBu) or methanol (for the blocks with PVDc and PVSt).
- Peak signals for the Tgs in DSC curves are very weak in Fig 7 and 8. I’m worried if the peak identification was good enough. Also, the authors simply state that 3 Tgs observed in the block copolymers are from each block and the microphase-separated structure. Authors need to provide supporting details for the microphase separation – or may be at least an appropriate reference.
The Δcp values of the glass transitions in the specific samples is very low and therefore, the transitions are generally weak. We verified the signals by close examination at the specific range of temperature values and repeated the measurements employing higher quantities of samples and also lower rates of heating (2 and 5oC/min) to increase the signals and verify their presence. A third transition in the block copolymers of this study was clearly shown for most of the samples indicating that the block copolymers are located to the weak segregation limit. This behavior is very well known in block copolymers (a reference was added in the text) and also has been reported for block copolymers containing PVEs (refs, 54, 55).
Reviewer 2 Report
Comments and Suggestions for Authors
Manuscript ID: polymers-3173587
Title: Block Copolymers of Poly(N-vinyl Pyrrolidone) and Poly(Vinyl Esters) bearing n-alkyl side-groups via RAFT polymerization. Synthesis, Characterization and Thermal Properties
This work is devoted to the synthesis and characterization of amphiphilic PNVP-b-PVEs block copolymers (NVP = N-vinyl pyrrolidone; VEs = different vinyl esters), as well as the study of their thermal properties. These polymeric materials have the numerous applications in the industrial sector and pharmacology. The synthesis of amphiphilic PNVP-b-PVEs block copolymers was carried out via the well-known Reversible Addition Fragmentation Chain Transfer (RAFT) methodology. The block copolymers were characterized by SEC, NMR spectroscopy, and their thermal properties were studied by DSC and TGA. Undoubtedly, this well-structured paper can be interesting to the readers of this journal. However, it should be improved prior to publication. Some comments are given below.
1) Abstract: “The thermal transitions of the copolymers were studied by…” should be written as “The thermal behavior of the copolymers were studied by…”.
2) Section 2.1: “N-Vinyl pyrrolidone (≥97% FLUCA)…” should be written as “N-Vinyl pyrrolidone (≥97% FLUKA)…”.
3) Section 2.1: The authors should provide the purity of vinyl esters (TCI Chemicals).
4) Table 1: “Quantities for the synthesis of the bock copolymers” should be written as “Quantities for the synthesis of the block copolymers”.
5) Table 2: copolymer PNVP-b-PVBu #2 has the highest values of Mn∙10³ = 28.0 and Mn∙10³ = 32.0 compared to other samples. The authors should comment this exceedance.
6) Figures 10, 12, 14: the Y axis should be named as “Weight loss (%)” (the weight loss vs. increasing temperature of the sample is recorded in the TGA experiments).
7) It is highly recommended to add the final (mass / mole) fraction purity values for all the synthesized copolymers (for example, as a new column in Table 1).
8) In context of the performed TGA (subsection 3.2.2), the authors should present schematically possible degradation processes for the studied copolymers.
Comments on the Quality of English Language
Minor editing of English language with checking some typos through the text required.
Author Response
We are thankful to the Reviewer for the fruitful comments and suggestions. Everything was taken into account and the manuscript was revised appropriately. The changes in the text are given by red color. Our point-by-point answers to the Reviewer are given below:
This work is devoted to the synthesis and characterization of amphiphilic PNVP-b-PVEs block copolymers (NVP = N-vinyl pyrrolidone; VEs = different vinyl esters), as well as the study of their thermal properties. These polymeric materials have the numerous applications in the industrial sector and pharmacology. The synthesis of amphiphilic PNVP-b-PVEs block copolymers was carried out via the well-known Reversible Addition Fragmentation Chain Transfer (RAFT) methodology. The block copolymers were characterized by SEC, NMR spectroscopy, and their thermal properties were studied by DSC and TGA. Undoubtedly, this well-structured paper can be interesting to the readers of this journal. However, it should be improved prior to publication. Some comments are given below.
- Abstract: “The thermal transitions of the copolymers were studied by…” should be written as “The thermal behaviorof the copolymers were studied by…”.
The expression was corrected in the text.
- Section 2.1: “N-Vinyl pyrrolidone (≥97% FLUCA)…” should be written as “N-Vinyl pyrrolidone (≥97% FLUKA)…”.
The expression was corrected in the text.
- Section 2.1: The authors should provide the purity of vinyl esters (TCI Chemicals).
The purity of the monomers was added in the text.
- Table 1: “Quantities for the synthesis of the bock copolymers” should be written as “Quantities for the synthesis of the blockcopolymers”.
The expression was corrected in the text.
- Table 2: copolymer PNVP-b-PVBu #2 has the highest values of Mn∙10³ = 28.0 and Mn∙10³ = 32.0 compared to other samples. The authors should comment this exceedance.
There is no specific reason for this difference in molecular weight for the present work. We just wanted to show that the higher molecular weight PNVP macro-CTA can be efficiently employed for the synthesis of copolymers with higher molecular weights. Recently, we synthesized more block copolymers and we are finishing our work studding the self-assembly behavior of the copolymers in THF (selective solvent for the PVEs blocks) and in aqueous solutions (selective solvent for the PNVP block). For this study we synthesized also higher molecular weight copolymers.
- Figures 10, 12, 14: the Y axis should be named as “Weight loss(%)” (the weight loss vs. increasing temperature of the sample is recorded in the TGA experiments).
The axis title was corrected.
- It is highly recommended to add the final (mass / mole) fraction purity values for all the synthesized copolymers (for example, as a new column in Table 1).
The overall polymerization yield was added in Table 1.
8) In context of the performed TGA (subsection 3.2.2), the authors should present schematically possible degradation processes for the studied copolymers.
It is beyond the scope of the present work to go into mechanistic details referring to the degradation mechanism of the copolymers. Taking into account the literature data (Refs. 72 and 79) it seems that the PVEs degrade by first removing the side alkyl group generating the corresponding acid and PVA. More experiments and combination of techniques are needed to clarify this point.
Round 2
Reviewer 1 Report
Comments and Suggestions for Authors
The authors have made effort to address the raised comments and there are no more open questions from my side
Comments on the Quality of English Language
-